# The Diagnosis of Wilkie’s Syndrome Associated with Nutcracker Syndrome: A Case Report and Literature Review

**DOI:** 10.3390/diagnostics14171844

**Published:** 2024-08-23

**Authors:** Ludovico Abenavoli, Felice Imoletti, Giuseppe Quero, Valentina Bottino, Viviana Facciolo, Giuseppe Guido Maria Scarlata, Francesco Luzza, Domenico Laganà

**Affiliations:** 1Department of Health Sciences, University “Magna Græcia”, Viale Europa, 88100 Catanzaro, Italy; felice.imoletti@hotmail.it (F.I.); bottinovalentinabv@gmail.com (V.B.); giuseppeguidomaria.scarlata@unicz.it (G.G.M.S.); luzza@unicz.it (F.L.); 2Digestive Surgery Unit, Fondazione Policlinico Universitario “A. Gemelli” IRCCS, Università Cattolica del Sacro Cuore, 00168 Rome, Italy; giuseppe.quero@policlinicogemelli.it; 3Department of Experimental and Clinical Medicine, University “Magna Græcia”, Viale Europa, 88100 Catanzaro, Italy; viviana.facciolo@gmail.com (V.F.); domenico.lagana@unicz.it (D.L.)

**Keywords:** diagnosis, treatment, imaging, syndromes

## Abstract

Superior mesenteric artery (SMA) syndrome or Wilkie’s syndrome is a vascular compression disorder that causes the abnormal compression of the third portion of the duodenum by the SMA. It has a low incidence rate, which is higher in young women, and is rarely associated with the Nutcracker phenomenon: a condition of the compression of the left renal vein between the SMA and the aorta, which manifests as pain in the left flank and pelvis. Here, we report on the case of a 54-year-old woman with a history of repeated episodes of abdominal pain caused by the Nutcracker syndrome and Wilkie’s syndrome.

## 1. Introduction

Superior mesenteric artery (SMA) syndrome, also known as Wilkie’s syndrome, is a rare condition where the third portion of the duodenum is compressed between the abdominal aorta and the SMA due to a lack of retroperitoneal fat. Symptoms include nausea, vomiting, postprandial abdominal pain, bloating, heartburn, and reflux, which can mimic anorexia nervosa and functional dyspepsia. This leads to a reduced caloric intake, weight loss, and further duodenal compression, creating a vicious cycle. The incidence of Wilkie’s syndrome is estimated at 0.013–0.3%, often diagnosed via computed tomography (CT) scanning by evaluating the SMA–aorta angle [1,2]. Barium X-ray may show characteristic features such as the dilatation of the duodenum and delayed gastroduodenal transit [2,3]. Nutcracker syndrome (NCS) involves the extrinsic compression of the left renal vein (LRV) between the aorta and SMA, leading to impaired blood flow and congestion. It manifests as flank and abdominal pain, varicocele, fatigue, and orthostatic intolerance [4]. NCS is categorized into the anterior and posterior types, with the former being more common [5]. Prevalence is higher in females, though the exact rates are unknown [6]. Associated conditions include pancreatic neoplasms, retroperitoneal tumors, and reduced retroperitoneal fat [5]. Diagnosis involves Doppler ultrasound, CT, magnetic resonance imaging (MRI), and retrograde venography [7]. The coexistence of Wilkie’s syndrome and NCS complicates diagnosis and treatment. Both syndromes present nonspecific gastrointestinal symptoms, with complicated differential diagnosis. Imaging methods like CT, Doppler ultrasound, and MRI may not clearly differentiate between the two conditions due to overlapping anatomical abnormalities. Evaluating venous pressure gradients and flow velocities can also be challenging as Wilkie’s syndrome may alter abdominal hemodynamics. Treatment strategies differ, with Wilkie’s syndrome often requiring nutritional support and NCS needing interventions for venous compression, making a comprehensive treatment plan essential. Surgical options and postoperative management are complex, requiring careful coordination among multiple specialties to ensure optimal outcomes [8]. Here, we report on the case of a patient with Wilkie’s syndrome combined with Nutcracker syndrome, with unusual presentation.

## 2. Case Report

This is the case of a 54-year-old woman with a body mass index of 23.2 kg/m^2^ and a history of sub-continuous epigastric pain, not related to meals, radiating to the back, with associated retrosternal heartburn, nausea, and frequent belching for which she took antacids and prokinetics, with partial benefit. She underwent an esophagogastroduodenoscopy (GIF 30-165, Olympus Medical Systems, Milan, Italy) to have two hyperplastic gastric polyps removed. In addition, a diagnosis of diffuse atrophic gastritis with high-titer anti-parietal cell antibodies (1:640) was made. Subsequently, she underwent a cholangio-MRI (Intera 1.5T, Philips Healthcare, Naples, Italy) with no evidence of the dilation of the intra- and extra-hepatic bile ducts or endoluminal stones. However, due to the persistence of symptoms, a cholecystectomy was performed. Postoperatively, the patient presented recurrent episodes of vomiting, and an abdominal CT scan (Aquilon 64, Toshiba, Mezzago, Italy) showed a fluid overdistension of the stomach and abdominal adhesions, leading to a viscerolytic surgery. However, after discharge, abdominal discomfort persisted, though more mildly, and the patient was admitted to our center. A complete blood count with no evidence of laboratory parameter alterations and, subsequently, an upper gastrointestinal (UGI) series (Omnidiagnost Eleva, Philips Healthcare, Naples, Italy) and a contrast-enhanced abdominal CT with contrast medium Iomeron 400 (Bracco S.p.a., Milan, Italy) were performed. The UGI series showed a slowed-down progression of the contrast medium, which stagnated at the level of the third duodenal portion (Figure 1).

The diagnosis of Wilkie’s syndrome (Figure 2) and NCS (Figure 3) was further confirmed via CT scan, with evidence of a significant restriction of the third duodenal portion as well as a compression with a proximal dilatation of the LRV.

Subsequently, the patient started a fodmap-free diet in combination with a treatment of trimebutyn maleate 150 mg thrice a day, simethicone 80 mg twice a day, and probiotic tablets once a day. This resulted in an improvement but not a reversal of the symptomatology due to the patient’s poor therapeutic and alimentary compliance. Currently, the patient is in clinical follow-up after a new endoscopic dilatation treatment. However, in accordance with the surgical team and the patient, a radical approach by gastrectomy is scheduled, in the case of the recurrence of abdominal symptoms.

## 3. Discussion

Considering the potentially associated life-threatening complications such as acute and chronic pancreatitis, severe malnutrition, duodenal and gastric ulcers, pneumoperitoneum and pneumomediastinum, Wilkie’s syndrome and NCS should always be taken into consideration in differential diagnosis in the case of patients with obstructive bowel symptoms. Both syndromes, although well described in the literature, are rare clinical pictures, usually presenting alone, and their coexistence is rare. Evidence described in the literature is summarized in Table 1.

In summary, the evidence indicates that the average age at diagnosis for patients with both syndromes was 28.72 years, with a standard deviation of 12.85 years. Additionally, there was a higher prevalence of males, comprising 54% (12/22) of the patients. Furthermore, these patients showed one to five clinical manifestations associated with both conditions. Regarding the patients’ management, CT scanning was the most used technique for diagnosis (16/22; 72%: Figure 4A), while nutritional therapy was the preferred treatment (11/22; 50%; Figure 4B).

The coexistence of Wilkie’s syndrome and NCS presents significant challenges in both diagnosis and treatment. Although US is proposed as a diagnostic exam, poor patient cooperation and intestinal meteorism may hinder the accurate viewing of abdominal structures. Using echo-color-Doppler modality, US can show bright colors of high velocity due to aliasing artifacts from the jetting flow distal to the aortomesenteric portion of the LRV. Its ability to visualize the acute angle of the SMA and detect high-velocity flows in the LRV makes US a powerful tool in the early stages of diagnosis. Additionally, the lower cost and higher accessibility of US compared to CT and MRI can facilitate more frequent and widespread screening, leading to the earlier detection and treatment of these syndromes. However, in our opinion, definitive diagnosis for both Wilkie’s syndrome and NCS is made with contrast-enhanced CT scanning or MRI, which detect duodenal and LRV compression and measure the SMA–aorta angle accurately [31,32]. The management of these patients is notably complicated, as highlighted by our patient who underwent a cholecystectomy for suspected symptomatic cholelithiasis before the actual diagnosis was reached. This initial misdiagnosis underscores the difficulty in distinguishing between the symptoms of Wilkie’s syndrome and other common gastrointestinal conditions. It also exemplifies the need for a multidisciplinary approach, involving gastroenterologists, radiologists, and surgeons, to accurately diagnose and treat these overlapping conditions. A detailed patient history, thorough physical examination, and appropriate imaging studies are crucial in avoiding such diagnostic pitfalls [25]. However, US can be used to identify the acute angle of the SMA relative to the aorta as a screening tool. It is not necessary to rely solely on the turbulence of the renal vein, although that can be easily observed. Conservative treatment is the most widely adopted for both conditions due to the associated benefits and lack of complications. However, surgery becomes necessary in cases where symptoms persist despite conservative strategies. Specifically, for Wilkie’s syndrome, surgery is indicated after the failure of conservative treatment, long-standing disease with progressive weight loss and duodenal dilatation with stasis, or complicated peptic ulcer secondary to bile stasis and reflux [33]. Surgical options include Strong’s procedure, gastrojejunostomy, and duodenojejunostomy, with the latter being the most successful at a rate higher than 90%, compared to Strong’s procedure at 25% [34]. For NCS, conservative treatment is preferred in young subjects due to the higher likelihood of spontaneous remission with increasing retroperitoneal fat and collateral venous circulation formation [35]. Surgical interventions are reserved for intolerable symptoms or the failure of conservative treatment, with LRV transposition being the most common and effective approach among various surgical options [35,36]. The complexity of managing patients with both syndromes necessitates a multidisciplinary approach [22]. In our patient’s case, the involvement of multiple specialties was crucial in eventually reaching the correct diagnosis: gastroenterologists were essential for managing gastrointestinal symptoms and nutritional support, while radiologists played a key role in accurately interpreting imaging studies. At the same time, surgeons were involved in both diagnostic laparoscopies and therapeutic interventions, such as the cholecystectomy and any subsequent surgical treatments for both conditions. Furthermore, we performed an updated review of published data, evaluating the pool of data emerging from the scientific literature. Our comprehensive analysis emphasizes the rarity and complexity of these conditions, the necessity of a thorough diagnostic workup, and the importance of a tailored treatment plan involving multiple medical disciplines. Recent studies have provided insights into the pathophysiology, diagnostic criteria, and treatment outcomes of both conditions, highlighting the need for continued research and clinical awareness. The clinical implications of patients’ management are crucial. Early and accurate diagnosis can prevent unnecessary surgical interventions, as shown in our patient who initially underwent a cholecystectomy. Furthermore, understanding the interplay between these syndromes can guide more effective treatment strategies. For instance, addressing nutritional deficiencies and weight loss can mitigate the symptoms of Wilkie’s syndrome, while appropriate surgical interventions can alleviate the vascular compression shown in NCS [24]. Furthermore, a better utilization of US in the diagnostic process can significantly add value to the literature by suggesting a more effective and cost-efficient approach. This can reduce the reliance on more expensive and invasive imaging techniques, ultimately improving patient care. In this way, the widespread use of US can lead to the development of more refined diagnostic criteria and protocols, enhancing the overall understanding and management of both syndromes [8,32]. Advancements in diagnostic imaging techniques and the integration of artificial intelligence (AI) hold promising potential for improving the accuracy and efficiency of diagnosing Wilkie’s syndrome and NCS. High-resolution imaging modalities, such as three-dimensional CT angiography and functional MRI, can provide a more detailed visualization of vascular and duodenal structures, enhancing the ability to detect subtle abnormalities and anatomical variations [37]. Moreover, AI-driven algorithms can assist in the interpretation of complex imaging data, reducing the likelihood of misdiagnosis. Machine learning models trained on large datasets could potentially identify patterns and markers specific to both conditions, facilitating early and accurate diagnosis. AI applications in imaging could also streamline the diagnostic workflow, enabling quicker decision-making and personalized treatment planning [38]. In addition to imaging advancements, the development of non-invasive biomarkers for these syndromes could further aid in diagnosis and monitoring. Research into the molecular and genetic underpinnings of Wilkie’s syndrome and NCS may uncover specific biomarkers detectable through blood tests or other minimally invasive methods, providing a complementary tool to imaging studies [27]. Future advancements in imaging technology and AI integration hold the potential to significantly enhance diagnostic accuracy and patient outcomes. Continued research and innovation in these areas are essential for developing more precise and personalized management strategies for patients with these complex and rare conditions. At the same time, observational studies on large cohorts of patients are necessary to define appropriate guidelines on the subject of diagnosis and to describe the possible treatment options to be adopted based on different clinical picture.

## 4. Conclusions

This case report explores the diagnostic and therapeutic challenges of coexisting Wilkie’s syndrome and NCS, conditions typically analyzed separately. Their simultaneous presence complicates diagnosis, highlighting the need to consider both in differential diagnosis. The patient in this report experienced persistent gastrointestinal symptoms, leading to a final diagnosis via advanced imaging methods like CT and MRI, which are crucial for accurate diagnosis. Conservative management, including dietary changes and medication, is often the first approach, but its success depends on patient compliance. When conservative treatment fails, surgical options such as duodenojejunostomy for Wilkie’s syndrome and LRV transposition for NCS can be effective. The decision for surgery requires careful consideration, and this report emphasizes the need for further research to develop comprehensive treatment guidelines.

## Figures and Tables

**Figure 1 diagnostics-14-01844-f001:**
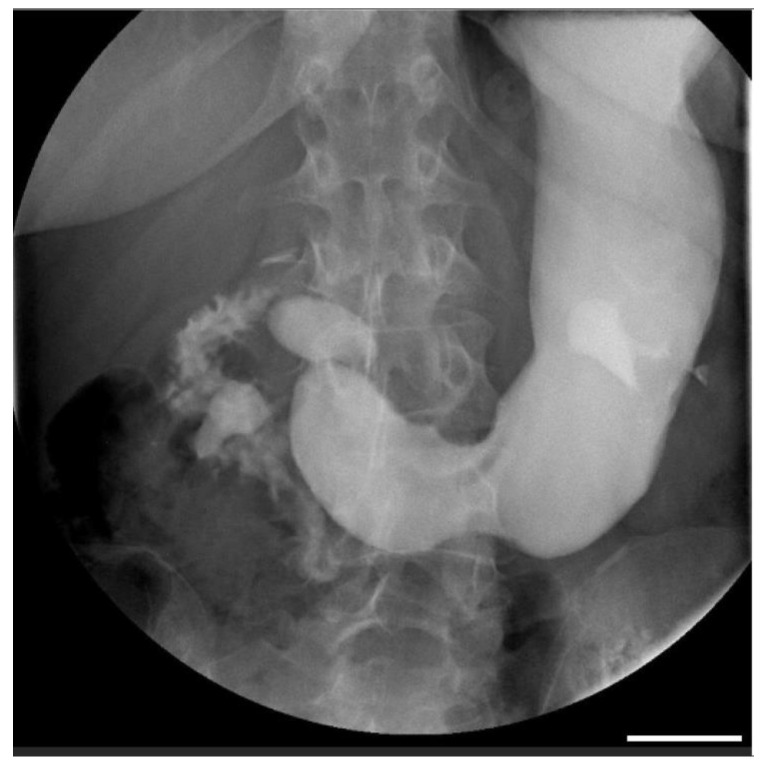
The antero-posterior projection of the UGI series showed the dilation of the stomach and second portion of the duodenum (scale bar: 1.5 cm).

**Figure 2 diagnostics-14-01844-f002:**
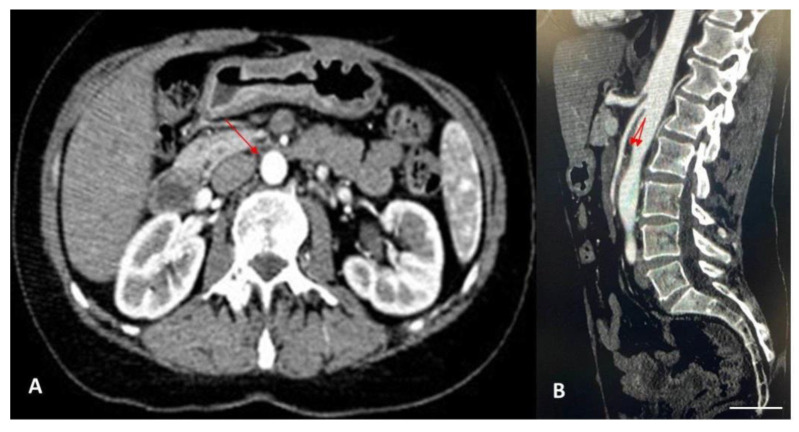
(**A**) Contrast-enhanced CT scan in axial projection showed the narrowing of the third duodenal portion between the aorta and the SMA (scale bar: 1 mm), as indicated by the red arrow (aorto-mesenteric distance of approximately 3 mm). (**B**) Contrast-enhanced CT scan in sagittal projection showed the narrowing of the third duodenal portion between the aorta and the SMA, with a reduction in the aorto-mesenteric angle of approximately 13° (scale bar: 1 mm), as indicated by the red arrow (normal values: 28–65°).

**Figure 3 diagnostics-14-01844-f003:**
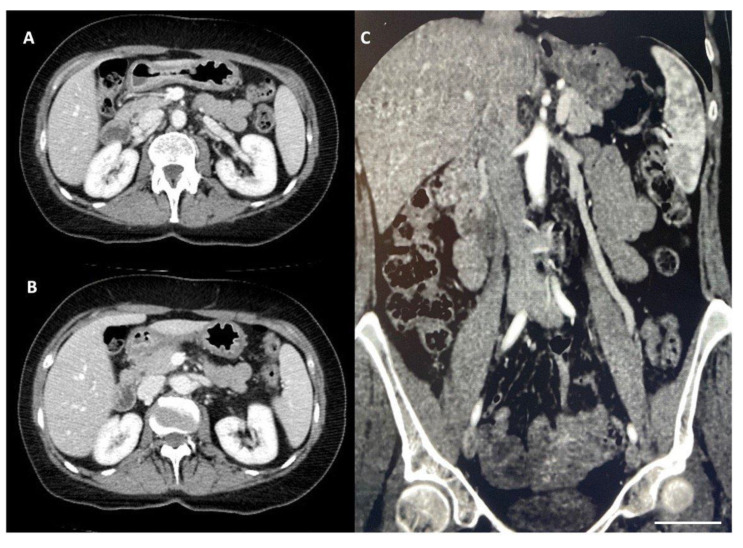
(**A**,**B**) Contrast-enhanced CT scan, in axial projection, showed the compression of the LRV (scale bar: 1 mm). (**C**) Reversal flow from the left gonadal vein in a coronal projection (scale bar: 1 mm).

**Figure 4 diagnostics-14-01844-f004:**
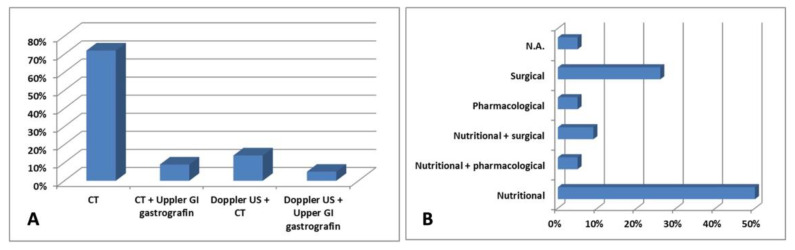
Different (**A**) diagnostic approaches and (**B**) treatment according to the studies in the literature.

**Table 1 diagnostics-14-01844-t001:** A summary of the different studies regarding the coexistence of Wilkie’s syndrome and NCS.

Reference	Patient	Clinical Manifestations	Diagnosis	Treatment
Barsoum et al., 2008 [9]	29-year-old female	Early satiety and postprandial epigastric abdominal pain	CT Upper GI gastrografin	Enteral nutrition
Vulliamy et al., 2013 [10]	55-year-old male	Vomiting, epigastric pain, and bloating	CT	N.A.
Inal et al., 2014 [11]	28-year-old male	Cachexia and intermittent abdominal pain	CT	Enteral nutrition
Alenezy et al., 2014 [12]	17-year-old male	Abdominal pain and intermittent vomiting	CT	Fluid and electrolyte replacement and nasogastric tube decompression
Nunn et al., 2015 [13]	19-year-old female	Severe epigastric pain associated with emesis and anorexia	CT	Enteral nutrition
Iqbal et al., 2016 [14]	62-year-old male	Cachexia	CT	Enteral nutrition
Heidbreder, 2018 [15]	20-year-old female	Severe left flank and lower left quadrant pain, abdominal pain, nausea, and vomiting	CT	Roux-en-Y duodenojejunostomy and LRV transposition
Al-Zoubi, 2019 [16]	38-year-old female	Intermittent left-sided loin pain	CT	LRV transposition
Shi et al., 2019 [17]	32-year-old female	Severe bloating, epigastric pain, left flank ache, nausea, and occasional vomiting	CT	Fluid resuscitation with parenteral and enteral nutritional support, plus mosapride citrate dispersible tablets 5 mg thrice a day
Diab et al., 2020 [18]	18-year-old male	Crampy postprandial abdominal pain associated with bilious vomiting, and signs of varicocele	CT	Regular assumption of a liquid diet
Lin et al., 2020 [19]	15-year-old male	Postprandial discomfort, nausea, and vomiting	CT	Enteral and parenteral nutrition
Farina et al., 2020 [20]	27-year-old male	Painful postprandial crises at the sub-acute onset, located at the epigastrium	Doppler US CT	Endovascular stent grafting
Wang et al., 2021 [21]	15-year-old male	Hematuria, fatigue, anorexia, nausea, and recurrent abdominal distension	Doppler US Upper GI gastrografin	Pulse dose of methylprednisolone 500 mg daily for 3 days, followed by 1 mg/kg orally and mycophenolate mofetil 0.75 g twice a day
Suarez-Correa et al., 2022 [22]	25-year-old male	Postprandial abdominal pain and distension, nausea, vomiting, and distension	CT Upper GI gastrografin	Enteral nutrition and surgery
Laskowski et al., 2022 [23]	40-year-old female	Nausea, early satiety, and diffuse abdominal pain	CT	LRV transposition
Khan et al., 2022 [24]	25-year-old female	Abdominal pain associated with nausea, bilious emesis, and diarrhea	CT	Surgery and conservative therapy
Ober et al., 2022 [25]	45-year-old female	Macroscopic hematuria, intermittent pain in the left flank and hypogastric region, postprandial nausea, and cachexia	Doppler US CT	Stent implantation in the LRV
Gungorer et al., 2022 [26]	17-year-old male	Abdominal pain, nausea, and vomiting	Doppler US CT	Surgery
Castro et al., 2023 [27]	18-year-old female	Epigastric pain and emesis	CT	Dietary changes
Pacheco et al., 2023 [28]	26-year-old male	GI obstructive symptoms	CT	Enteral nutrition
Alonso-Canal et al., 2023 [29]	24-year-old male	Functional dyspepsia	CT	Dietary changes
Brogna et al., 2023 [30]	37-year-old female	Abdominal pain with sub-occlusive episodes, nausea, and vomiting	CT	Periodic insertion of a nasogastric tube to decompress the stomach, along with a high-protein diet and parenteral nutritional supplements

Abbreviations: CT: computed tomography; GI: gastrointestinal; N.A: not available; LRV: left renal vein; US: ultrasound.

## Data Availability

All of the data are included in this study.

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
