# Peer review of "The Diagnosis of Wilkie’s Syndrome Associated with Nutcracker Syndrome: A Case Report and Literature Review"

_diagnostics, 2024, doi:10.3390/diagnostics14171844_

Round 1
Reviewer 1 Report
Comments and Suggestions for Authors
I am really grateful to review this manuscript. In my opinion, this manuscript can be published in current form. This study reported a case of a 54-year-old female patient with Wilkie’s syndrome associated with Nutcracker 2 syndrome. The diagnosis of Wilkie’s syndrome associated with Nutcracker 2 syndrome was further confirmed via computed tomography (CT) scan, with evidence of a significant restriction of the third duodenal portion as well as a compression with proximal dilatation of the left renal vein. Indeed, this study presented the summary of existing evidence: the mean age 28.72, the male prevalence 54%, the CT scan (or nutritional therapy) as the approach of diagnosis 72% (or 50%). I would argue that this is a good achievement. Also, I would like to express my strong expectation regarding its follow-up studies.
Comments on the Quality of English LanguageMinor editing of English language required
Author Response
I am really grateful to review this manuscript. In my opinion, this manuscript can be published in current form. This study reported a case of a 54-year-old female patient with Wilkie’s syndrome associated with Nutcracker 2 syndrome. The diagnosis of Wilkie’s syndrome associated with Nutcracker 2 syndrome was further confirmed via computed tomography (CT) scan, with evidence of a significant restriction of the third duodenal portion as well as a compression with proximal dilatation of the left renal vein. Indeed, this study presented the summary of existing evidence: the mean age 28.72, the male prevalence 54%, the CT scan (or nutritional therapy) as the approach of diagnosis 72% (or 50%). I would argue that this is a good achievement. Also, I would like to express my strong expectation regarding its follow-up studies.
Reply. Thank you for your positive comments.
Reviewer 2 Report
Comments and Suggestions for Authors
Dear colleagues.
Thank you for your submission.
I have few comments:
- Introduction is too extensive for the presentation of a case report in my opinion. It should just highlight the main controversies and areas of improvement in such pathology.
- You should highlight the special characteristics in your case in order to make it special to be published. Otherwise, you are not contributing nothing new to the current knowledge.
- Taking into consideration the topic of the special issue to which you are submitting the manuscript, I would expect to find in your manuscript any new approach for diagnosis or at least any look into future possible perspectives, but I did not find any of them. That can be improved.
- Although your are trying to focus in the diagnosis, it seems that the case report is not concluded as the definitive treatment has not been still done. Perhaps it is more useful to present it once it is completely solved.
Comments on the Quality of English Language
Author Response
Dear colleagues.
Thank you for your submission.
I have few comments:
- Introduction is too extensive for the presentation of a case report in my opinion. It should just highlight the main controversies and areas of improvement in such pathology.
Reply 1. Thank you for your comment. The text has been revised (see lines 23-51).
- You should highlight the special characteristics in your case in order to make it special to be published. Otherwise, you are not contributing nothing new to the current knowledge.
Reply 2. Thank you for your comment. The text has been revised (see lines 148-160).
- Taking into consideration the topic of the special issue to which you are submitting the manuscript, I would expect to find in your manuscript any new approach for diagnosis or at least any look into future possible perspectives, but I did not find any of them. That can be improved.
Reply 3. Thank you for your comment. Our case does not report any innovative diagnostic approach as this was performed following the indications from the scientific literature. However, future perspectives have been added to the conclusion section (see lines 181-200).
- Although your are trying to focus in the diagnosis, it seems that the case report is not concluded as the definitive treatment has not been still done. Perhaps it is more useful to present it once it is completely solved.
Reply 4. Thank you for your comment. The text has been revised (see lines 88-91).
Reviewer 3 Report
Comments and Suggestions for Authors
Dear Authors,
Good literature review of this rare condition. Very well written. Line 124, mean needs to be corrected.
Author Response
Dear Authors,
Good literature review of this rare condition. Very well written. Line 124, mean needs to be corrected.
Reply. Thank you for your positive comments. The text has been revised (see lines 104-106).
Round 2
Reviewer 2 Report
Comments and Suggestions for Authors
Dear authors.
Thank you for the effort than to make the proposed changes.
I still have two comments:
- The first one is a general one, as I consider the conclusions too long, mostly for a case report. Regarding future perspectives, I would suggest to include them as part of the discussion, not the conclusions section.
- The second one is that I cannot see anything that I don't see anything that would make it an article worth including in the special issue of Advances in the Diagnostic Imaging of Gastrointestinal Diseases, as it doesn't seem to me that the diagnostic methods used are any novelty but simply that the infrequency of the case is striking.
Author Response
Thank you for the effort than to make the proposed changes.
I still have two comments:
- The first one is a general one, as I consider the conclusions too long, mostly for a case report. Regarding future perspectives, I would suggest to include them as part of the discussion, not the conclusions section.
Reply 1. Thank you for your comment. The text has been revised (see lines 172-191; 195-206).
- The second one is that I cannot see anything that I don't see anything that would make it an article worth including in the special issue of Advances in the Diagnostic Imaging of Gastrointestinal Diseases, as it doesn't seem to me that the diagnostic methods used are any novelty but simply that the infrequency of the case is striking.
Reply 2. Thank you for your comment. The text has been revised (see lines 119-122; 167-172).